# Differential Effects of Exercise on fMRI of the Midbrain Ascending Arousal Network Nuclei in Myalgic Encephalomyelitis/Chronic Fatigue Syndrome (ME/CFS) and Gulf War Illness (GWI) in a Model of Postexertional Malaise (PEM)

**DOI:** 10.3390/brainsci12010078

**Published:** 2022-01-05

**Authors:** James N. Baraniuk, Alison Amar, Haris Pepermitwala, Stuart D. Washington

**Affiliations:** Department of Medicine, Georgetown University, Washington, DC 20007, USA; aca117@georgetgown.edu (A.A.); hsp9@georgetown.edu (H.P.); sdw4@georgetown.edu (S.D.W.)

**Keywords:** midbrain, postexertional malaise, PEM, arousal, exercise, fMRI, autonomic, postural tachycardia, Myalgic Encephalomyelitis/Chronic Fatigue Syndrome, ME/CFS, Gulf War Illness, GWI

## Abstract

Background: Myalgic Encephalomyelitis/Chronic Fatigue Syndrome (ME/CFS), Gulf War Illness (GWI) and control subjects underwent fMRI during difficult cognitive tests performed before and after submaximal exercise provocation (Washington 2020). Exercise caused increased activation in ME/CFS but decreased activation for GWI in the dorsal midbrain, left Rolandic operculum and right middle insula. Midbrain and isthmus nuclei participate in threat assessment, attention, cognition, mood, pain, sleep, and autonomic dysfunction. Methods: Activated midbrain nuclei were inferred by a re-analysis of data from 31 control, 36 ME/CFS and 78 GWI subjects using a seed region approach and the Harvard Ascending Arousal Network. Results: Before exercise, control and GWI subjects showed greater activation during cognition than ME/CFS in the left pedunculotegmental nucleus. Post exercise, ME/CFS subjects showed greater activation than GWI ones for midline periaqueductal gray, dorsal and median raphe, and right midbrain reticular formation, parabrachial complex and locus coeruleus. The change between days (delta) was positive for ME/CFS but negative for GWI, indicating reciprocal patterns of activation. The controls had no changes. Conclusions: Exercise caused the opposite effects with increased activation in ME/CFS but decreased activation in GWI, indicating different pathophysiological responses to exertion and mechanisms of disease. Midbrain and isthmus nuclei contribute to postexertional malaise in ME/CFS and GWI.

## 1. Introduction

Myalgic Encephalomyelitis/Chronic Fatigue Syndrome (ME/CFS) [1,2] and Gulf War Illness (GWI) [3,4] share features of postexertional malaise (PEM, exertional exhaustion), fatigue that is not relieved by rest, unrefreshing and non-restorative sleep, total body pain and systemic hyperalgesia. ME/CFS has been considered to be a chronic consequence following flu-like epidemics [5], but in general has a sporadic heterogeneous presentation and unknown etiology. The prevalence is about 0.2 to 2% [6,7]. The 1994 Center for Disease Control criteria (“Fukuda”) require moderate-to-severe unremitting fatigue of new onset that persists for longer than 6 months and has no explanation despite appropriate medical investigations and at least four of the following eight ancillary criteria: cognitive complaints regarding short-term memory or concentration, sore throat, sore lymph nodes, myalgia, arthralgia, headaches (including migraine), disordered sleep and postexertional malaise (PEM) [1]. Emphasis has been placed on PEM, the characteristically delayed exacerbation of the entire symptom complex following minimal physical, cognitive or emotional efforts, as a distinguishing feature of ME/CFS [3,4,7,8].

GWI affects 25 to 32% of veterans deployed in the 1990–1991 Persian Gulf War. The Centers for Disease Control criteria for Chronic Multisymptom Illness (CMI) require symptoms from at least two of three clusters: general fatigue; mood and cognitive abnormalities; and myalgia/arthralgia (pain) [3]. Mood/cognitive symptoms can range from troubles with sleep to cognitive difficulties, anxiety and depressive mood. An epidemiological comparison of symptoms between deployed and non-deployed Kansas veterans generated a more sensitive set of criteria requiring three of the following six domains: fatigue and sleep; pain; neurological/cognitive/mood; gastrointestinal; respiratory; and skin symptoms [4]. The etiology has been linked to exposures to neurotoxicants that were present in theatre, including organophosphates, carbamates and other pesticides, sarin/cyclosarin nerve agents and pyridostigmine bromide used as prophylactic medication against chemical warfare attacks [9,10]. Symptoms are consistent with Chronic Organophosphate-Induced Neuropsychiatric Disorder (COPIND) [11,12,13]. Psychiatric etiologies have been ruled out [9].

The overlapping symptoms of GWI and ME/CFS have generated unified hypotheses [10]. However, pathophysiological mechanisms and objective findings that can be used for disease diagnosis and the prediction of potential therapies have been more difficult to identify. We approached this problem by developing a two-day submaximal exercise provocation paradigm with symptom, heart rate variability, and functional magnetic resonance imaging to assess PEM and other changes induced by the exertional challenge. In a previous study, we examined blood oxygenation level-dependent (BOLD) activation during a difficult, high-cognitive-load continuous 2-back working memory task and compared pre-exercise and postexercise scans. The control, ME/CFS and GWI groups were equivalent prior to exercise (baseline), but after exercise ME/CFS subjects had a significant increase in blood oxygenation level-dependent (BOLD) activation while GWI participants had a significant decrease in the dorsal midbrain, right middle insula and left Rolandic operculum [14]. The midbrain region of interest extended from the left to right periaqueductal gray (PAG) and to the adjacent right midbrain reticular formation (MRF), inferior colliculus and lateral lemniscus, and caudally to the right lateral isthmus (Appendix A). Because these nuclei have profound influences on threat assessment, pain, negative emotion, attention, wakefulness, and instinctual neurobehaviors, it was of interest to assess the activation of relevant anatomical midbrain nuclei.

In this report, we re-analyze the original BOLD data [14] (Appendix A) using a seed region approach to gain a preliminary understanding of the nuclei that were activated within the midbrain region of interest. The seed regions were selected from the ascending arousal network that was defined from histological sections and diffusion studies of brainstem white matter tracts [14]. BOLD signals in each of the target nuclei were assessed on pre-exercise and postexercise days. The aim was to identify which midbrain nuclei were affected by exercise in ME/CFS and GWI subjects and to judge effect sizes to guide future confirmatory studies. We propose that affected nuclei may participate in the pathology of postexertional malaise.

## 2. Methods

### 2.1. Ethics

All subjects gave written informed consent to this protocol that was approved by the Georgetown University Institutional Review Board (IRB 2009-229, 2013-0943 and 2015-0579) and Department of Defense Congressionally Directed Medical Research Program (CDMRP) Human Research Program Office (HRPO) (A-15547 and A-18479) and listed in https://clinicaltrials.gov (accessed on 21 November 2021) (NCT01291758 and NCT00810225). All clinical investigations were conducted according to the principles expressed in the Declaration of Helsinki.

### 2.2. Demographics

GWI, ME/CFS and healthy control subjects were recruited to this four-day-long in-patient study in the Clinical Research Unit of the Georgetown—Howard Universities Center for Clinical and Translational Science. Subjects had history and physical examinations to ensure their inclusion by meeting Chronic Multisymptom Illness [3] and Kansas [4] criteria for GWI, Fukuda [1] and Canadian [2] criteria for ME/CFS, confirmation of sedentary lifestyle for control subjects (less than 40 min of aerobic activity per week) and exclusion because of serious medical or psychiatric conditions such as psychosis [4,15,16,17]. History of post-traumatic stress disorder (PTSD) [18] or depression [19] were not exclusions unless the subject had been hospitalized in the past 5 years. Subjects completed the Chronic Fatigue Syndrome Symptom Severity [20], SF-36 quality of life [21], Chalder Fatigue [22] and McGill Pain [23] questionnaires and had systemic hyperalgesia tested by dolorimetry [24,25].

Light sensitivity was assessed by comparing scores while looking up at standard hospital ward fluorescent lights turned on or off with curtains shut and background ambient light in the supine position. Sound sensitivity was assessed by dropping a standard hardboard clipboard (#44292, Item #1671406, Model #ST44292-CC, Staples, Inc. Framingham, Massachusetts 01702-4478, U.S.A.) from 3 feet onto a linoleum floor. The sound intensity 6 feet away at the subject’s ear was approximately 60 dB (ambient noise ~35 dB) (Physics Toolbox Sensor Suite). Severity was scored using the modified 0-to-20-point anchored ordinal intensity scale of the Gracely Box score [26,27,28].

### 2.3. Exercise Provocation

Two submaximal bicycle exercise tests were performed 24 h apart. Subjects cycled for 25 min at 70% of their predicted maximum HR (220 minus patient’s age), followed by a climb to 85% maximum HR to reach their anaerobic threshold [14,26,27,29,30]. Identical magnetic resonance imaging scans were run before the first and after the second stress tests, which were then contrasted to show the effects of exertion on cognitive tasks.

### 2.4. Orthostatic Postural Tachycardia Phenotypes

Subjects rested in the supine position for 5 min with continuous measurements of EKG and arm cuff blood pressure every minute. Average recumbent heart rate (HR) was calculated. Subjects stood up and maintained their posture for 5 min. EKG and blood pressure were recorded each minute. The differences between standing HR at each minute and average recumbent HR were calculated (∆HR). The procedure was performed at least twice before the first exercise, then 1, 3, 8 and 24 h post exercise. ∆HR was used to define Orthostatic status [26,27].

(i)Postural orthostatic tachycardia (POTS) was defined by ∆HR ≥ 30 beats per minute at least 4 time points before exercise and during each postexercise measurement period.(ii)Stress Test Activated Reversible Tachycardia (START) was defined by a normal ∆HR before exercise, but at least 2 episodes with ∆HR ≥ 30 beats per minute after exercise. The phenomenon was transient as postural tachycardia returned to normal within 36 to 48 h.(iii)The normal postural response was defined as Stress Test Originated Phantom Perception (STOPP) based on original findings in GWI [30,31] where ∆HR was in the normal range of 12 ± 5 beats per minute and never exceeded 30 beats per minute.

### 2.5. Verbal Working Memory Task

Subjects practiced the 0-back and 2-back working memory task in a mock scanner until they felt proficient [14,30,31]. In the scanner, subjects viewed an instruction panel stating “REST” for 0.8 s, followed by 19.2 s of a blank screen. The instruction “0-BACK” was viewed for 0.8 s, followed by 1.2 s of a blank screen, and then a string of nine pseudorandomized letters (A, B, C, D) were seen for 0.8 s each, followed by 1.2 s of a blank screen per letter. Each time they saw a letter, subjects pressed the corresponding button on an MRI compatible fiberoptic four-button box that was used with both hands. After a second “REST” period, they saw the instruction “2-BACK” and again viewed a string of nine letters. They had to view and remember the first two letters, then press the button for the first letter when they saw the third letter (i.e., “2 back”, 4 s delay). The 2-back task continued for seven responses. This cycle was repeated five times.

### 2.6. MRI Data Acquisition, Preprocessing and Analysis

All structural and functional MRI data were acquired on a Siemens 3 T Tim Trio scanner located within the Center for Functional and Molecular Imaging at Georgetown University Medical Center equipped using a transmit–receive body coil and a commercial 12-element head coil array as described previously [14,30,31]. Parameters for structural 3D T1-weighted magnetization-prepared rapid acquisition with gradient echo (MPRAGE) images were: TE = 2.52 ms, TR = 1900 ms, TI = 900 ms, flip angle = 9, FOV = 250 mm, 176 slices, slice resolution = 1.0 mm and voxel size 1 × 1 × 1 mm. Images were processed in SPM12 [32]. fMRI data consisted of T2*-weighted gradient-echo planar images (EPIs) acquired during the n-back tasks. The EPI data acquisition parameters were: TR/TE = 2500/30 ms, flip angle = 90, FoV = 205 mm^2^, matrix size = 64 × 64, number of slices = 47, and voxel size = 3.2 mm^3^ (isotropic). Raw EPI data were preprocessed through the default pipeline within the CONN toolbox [33]. Briefly, steps were: (i) slice timing correction, (ii) subject motion estimation and correction, (iii) outlier detection for “scrubbing” based on Artifact Detection Tools, (iv) co-registration with structural data, (v) segmentation and spatial normalization into standard Montreal Neurological Institute (MNI) space [34] and (vi) spatial smoothing with a stationary Gaussian filter with a full width of 6 mm at half maximum (FWHM). The voxel size was 2.0 mm^3^ (isotropic) after spatial normalization and conversion to Montreal Neurological Institute (MNI) space.

All within-subject and group-level image analyses were performed using the SPM12 software package [35]. After accounting for magnetic saturation by removing the first 6 scans, a paradigm based on the timing of events in the 2-back task (Figure 1) was applied to preprocessed EPI data to sort individual subject scans into instruction, fixation, 0-back and 2-back bins. In the original analysis, one-sample *t*-tests contrasted the BOLD signals from the 2-back and 0-back scans of each subject and included estimates of the translation (x, y, and z) and rotation (roll, pitch, and yaw) as covariates of non-interest. The resulting 2-back>0-back contrast maps from every subject were sorted into the control, GWI, and ME/CFS groups [14].

For this seed region re-analysis, we used the Harvard Ascending Arousal Network atlas [36,37] in the Lead DBS software package [38,39,40] to define regions of interest (ROI) for midbrain nuclei. The mean BOLD signal for each ROI was extracted from each subject’s contrast map, re-centered to a population grand mean of 0, and the normalized data were analyzed in the MarsBaR 0.44 toolbox [41,42]. The MarsBaR output was the BOLD activation levels for the 2-back>0-back contrast condition in each midbrain nucleus for the control, ME/CFS and GWI groups on the pre- and postexercise days.

Significant differences were determined by analysis of variance (ANOVA) with Tukey Honest Significant Difference or 2-tailed unpaired *t*-test with Bonferroni correction to correct for multiple comparisons, multivariate general linear modeling (mGLM) of relevant demographic and other independent variables, and partial correlation analysis performed in R and SPSS v27. The mGLM of BOLD activation in each seed region began with disease status (control, ME/CFS, GWI), orthostatic status (START, STOPP, POTS), PTSD and gender as fixed factors with age, BMI and dolorimetry pressure thresholds as independent variables. The results are shown in the Appendix A.

Nuclei that were significantly altered based on ANOVA were annotated onto the DBS Harvard anatomical figure (Figure 1, Table 1) [36,37]. When viewed from the anterior (ventral) and posterior (dorsal) sides, the nuclei in the ascending arousal network suggest three layers in the coronal plane of MRI space that approximate their embryological origins [43,44,45]. Mesomere 1 contributes the superior (SC) and inferior colliculus (IC), midbrain reticular formation (MRF) and periaqueductal grey (PAG). The second layer is derived from rhombomeres 0 and 1 and includes the superior dorsal raphe (DR), pontis oralis (PO), and more lateral pedunculotegmental nuclei (PTN, formerly pedunculopontine nuclei or PPN) [43,44,45]. The ventral tegmental area (VTA) is anterior when viewed at this level but extends embryologically from the isthmus to diencephalon. The most caudal layer had midline DR and median raphe (MR), and bilateral locus coeruleus (LC) and parabrachial complex (PBC) that are derived from rhombomeres 1 and 2. Embryological origins do not align with the coronal MRI projection because of the marked ventral flexion of the midbrain during the development and distortion of the original neural tube structures. Nuclei with significant differences by BOLD were manually highlighted by red outlines using Procreate software.

## 3. Results

### 3.1. Demographics and Questionnaires

As expected, there were more females in the ME/CFS group because of the known female predominance [6] and more male veterans in the GWI group from assault divisions deployed to the Persian Gulf (Table 2). PTSD was more common in the GWI subjects. Pressure-induced pain sensitivity tested by dolorimetry was not different because male and female subjects were combined [25]. The ME/CFS and GWI groups had comparable symptom scores that were significantly worse than controls with the exception of pain, SF36 Role Emotional and Mental Health, which indicated more impairment in the GWI group than ME/CFS group.

Pre-exercise BOLD values were assessed by self-reported demographic variables in a multivariable general linear model. The significant covariates were Orthostatic status, low back pain, depression, heart disease, gender and marital status (Appendix A). The next iteration used the significant covariates as fixed factors and removed the other variables. Orthostatic status was the only variable to be significant (Appendix A).

### 3.2. Partial Correlations

Partial correlations compared BOLD signal intensities for each node on Day 1, Day 2 and the delta with subjective questionnaires about CFS symptoms, SF36 domains, psychological and depression complaints with disease status, orthostatic status, gender, age and BMI as covariates. BOLD data were internally correlated within Day 1, Day 2 and delta, positively correlated between Day 2 and delta, and negatively correlated for Day 1 vs. delta (Appendix A). There were no significant correlations between subjective questionnaire data and objective pre-exercise or postexercise BOLD outcomes. The magnitudes of the significant correlations (*p* < 0.05 corrected) were low (R < 0.4 and R > −0.4 for the inversely scored SF36 domains).

### 3.3. ANOVA

BOLD data were compared between groups defined by disease status (control, ME/CFS, GWI) and orthostatic status (START, STOPP, POTS) on the pre-exercise and postexercise study days. In the original study, dorsal midbrain activation was not different between groups before exercise [14].

A visual inspection suggested a trend for differences in BOLD between groups. Data from all seed region datapoints and subjects were contrasted between groups. Prior to exercise, the ME/CFS group had numerically lower BOLD values (0.108 ± 0.032, mean ± 95%CI for n = 504 datapoints = all regions of interest in all ME/CFS subjects) compared to controls (0.297 ± 0.037, mean ± 95%CI, n = 434 datapoints) and GWI subjects (0.235 ± 0.024, mean ± 95%CI, n = 1092 datapoints). This suggested reduced blood flow during the 2-back > 0-back condition for ME/CFS at baseline.

In the ascending arousal network, bilateral PTN, L_PBC, and VTA were significantly more activated in the control group than the ME/CFS group, while the GWI group was higher in L_PTN and L_PO than ME/CFS subjects (Figure 2, Table 3).

The only difference based on Orthostatic status on Day 1 was the greater L_PBC activation in POTS than STOPP (Appendix A).

Males (0.249 ± 0.023, mean ± 95%CI) showed a trend towards higher BOLD activation than females (0.167 ± 0.028, *p* = 0.000012 by unpaired two-tailed *t*-test) when all nodes and subjects were assessed. The difference was significant for L_PO (*p* = 0.046) (Appendix A).

On the postexercise day, the relationship between groups became inverted as the control (0.254 ± 0.035, mean ± 95%CI) and ME/CFS (0.260 ± 0.034) groups had significantly greater BOLD than GWI subjects by ANOVA when all subjects and nodes were evaluated (*p* < 10^−9^ by two-tailed unpaired *t*-test with Bonferroni correction). After exertion, the control group was higher in bilateral MRF, VTA and R_PTN than GWI, while ME/CFS subjects had greater midline PAG, DR and MR and right MRF, LC and PBC than GWI ones (Table 4) (Figure 3).

There were no significant differences based on Orthostatic status following exercise (Appendix A).

Gender was not a significant covariate for any region post exercise (Appendix A). However, there was a general trend for females (0.123 ± 0.028, mean ± 95%CI) to have lower BOLD than males (0.197 ± 0.021, *p* = 0.000032 by two-tailed unpaired *t*-test) when all nodes and subjects were compared.

ΔBOLD was positive in the ME/CFS group and negative in the GWI group, indicating the significant dynamic effects caused by exercise in these two diseases (Figure 4). ME/CFS had higher increments than GWI for all seed regions except MRF and L_LC (Table 5). There were no differences between groups defined by Orthostatic status or gender.

### 3.4. Multivariate General Linear Models (mGLM)

The mGLM of pre-exercise data used Disease, orthostatic status, PTSD and gender as fixed factors with age, BMI and dolorimetry pressure thresholds as independent variables. Disease status was significant (Appendix A). L_PTN activation was significantly lower in ME/CFS subjects (0.018 ± 0.143, mean ± 95%CI) than control (0.326 ± 0.198, *p* = 0.047 univariate significance) and GWI participants (0.286 ± 0.127, *p* = 0.018) (Appendix A). This was comparable to the ANOVA outcomes (Table 3). Orthostatic status was significant, with L_PBC being significantly lower in STOPP (0.062 ± 0.121, mean ± 95%CI) than POTS (0.394 ± 0.219, *p* = 0.006 Tukey Honest Significant Difference) and START (0.397 ± 0.164, *p* = 0.034) (Appendix A). Age, gender, PTSD, BMI and dolorimetry pressure thresholds were not significant covariates prior to exercise.

Postexercise mGLM evaluated the same fixed factors and independent variables. Disease status was significant after exercise. ME/CFS and control participants had significantly higher BOLD activation than GWI in VTA, L_MRF and R_PTN (Appendix A). Overall, ME/CFS was greater than GWI for all regions except L_PO, L_LC and L_PBC. More nodes were significant by mGLM than ANOVA (Table 4). Gender was significant for R_LC and R_PBC as males had greater BOLD activation than females after adjustment for the other variables (Appendix A). Other significant interactions between disease, orthostatic and PTSD status, age and dolorimetry thresholds (kg) were detailed in the Appendix A, but must be interpreted with caution because of concerns about the number of variables, sample sizes and potential overfitting of the data.

Incremental changes in BOLD between days (ΔBOLD) reinforced the differences between diseases found by ANOVA (Table 5). The estimated marginal means for disease status, bracketed zero for controls, were positive for ME/CFS and negative for GWI (Appendix A).

Orthostatic status had a significant impact with higher activity for STOPP than START in R_LCΔ and bilateral PBCΔ (Table 6). The 95% confidence intervals for POTS and STOPP bracketed zero, indicating no change after exercise. However, ΔBOLD was negative for the START group, indicating a dynamic exercise-induced effect on brainstem activation in this phenotype. Effect sizes were small, indicating that it may be difficult to reproduce the finding.

### 3.5. Light and Sound Sensitivity

Provocations with light and sound showed that sensory sensitivities were significantly worse in ME/CFS and GWI than control subjects before and after exercise (Figure 5). A frequency analysis of light sensitivity prior to exercise found scores of 0 or 1 out of 20 (no discomfort) in 83.3% of SC, 30.8% of ME/CFS and 21.1% of GWI groups. Exercise worsened light sensitivity in paired analysis for the CFS (*p* = 2.7 × 10^−6^) and GWI groups (*p* = 0.022), and sound sensitivity in the ME/CFS (*p* = 0.037) group by 2-tailed paired *t*-tests. The incremental changes (∆) were significantly larger in the ME/CFS group than the GWI and control groups (*p* < 0.044 by 2-tailed unpaired *t*-tests after Bonferroni corrections). Thresholds for significant sensitivities were ≥2 out of 20 by receiver operating characteristics, indicating that visual and auditory hypersensitivity was common in the ME/CFS and GWI groups. The sound sensitivity and exaggerated startle responses were consistent with dysfunctional activity in the inferior colliculus [14].

## 4. Discussion

The BOLD data are the 2-back>0-back condition that contrasts the difficult high-cognitive-load continuous 2-back working memory task against the simple low cognitive load 0-back stimulus matching attention task [14]. The postexercise and incremental data reflect the dynamic effects of exertion on cognition. Exercise caused changes in the 2-back>0-back condition that measures relative brain activation during the more difficult task. Specific effects on the 2-back alone, 0-back alone, and 0-back>2-back conditions were not assessed here.

The importance of the BOLD data is that there were differences in the relative levels of regional blood flow into midbrain nuclei. ME/CFS and GWI subjects had significant incremental changes following exercise whereas controls had no net changes. The changes seen in ME/CFS —and GWI subjects were antithetical to each other, indicating distinct dynamic exercise-induced pathological consequences.

In the original study [14], dorsal midbrain activation was not different between groups prior to exercise. A re-analysis of the same BOLD data using the ascending arousal network nuclei and seed region approach found baseline differences with lower BOLD in ME/CFS than control and GWI (Figure 2, Table 3). Controls had greater activation compared to ME/CFS in VTA, bilateral PTN and L PBC, while GWI subjects were higher than ME/CFS for left PTN and PO prior to exercise.

The pedunculotegmental nuclei (PTN) had reduced activation in ME/CFS. This cholinergic nucleus has extensive efferents that release acetylcholine throughout the cerebrum to maintain wakefulness and sustain attention. PTN assists in updating rapidly changing environmental information as required for our continuous 2-back task. Other functions and potential roles in disease dysfunction are discussed in the accompanying paper in this issue.

Exercise caused a significant dynamic switch in midbrain activation. Exercise caused an increase in BOLD in the ME/CFS group but a decrease in GWI subjects. Therefore, exercise had differential effects in the ME/CFS group compared to GWI subjects. Following exercise, the controls were greater than GWI subjects in VTA, bilateral MRF and right PTN, while the ME/CFS group was higher than GWI for midline PAG, DR and MR, and right lateral MRF, PBC and LC (Figure 3, Table 4). The seed region approach was consistent with the activation found in the original 141-voxel region of interest that also included inferior colliculus (Appendix A) [14]. All 141 voxels in the region of interest analysis were contiguous, but the seed region approach assayed the net activation within the smaller volumes defined by the seeds.

After exercise, the midline nuclei and MRF had significantly lower BOLD activation in GWI than the other two groups.

PAG is integral to threat assessment and instantaneous responses. The detection of a threat activates the PAG and midbrain reticular formation and causes a transition from relaxed wakefulness to high general attention [46]. Active responses range from freeze with conscious tonic immobility if motion would lead to detection by a nearby predator; a defensive approach to assess an ambiguous threat (modeled as rumination) [47,48]; flight via an escape route or shelter; and defensive attack if the predator is within a dangerous distance and escape is not possible [49,50].

VTA is a dopaminergic nucleus that stimulates the locus coeruleus to promote wakefulness [51,52]. It has a role in reward situations and positive emotion. Dysfunction is associated with anhedonia.

DR and MR are serotoninergic nuclei that project to the limbic system during active stress, conflicts and anxiety [53,54]. They may initiate fight or flight decisions. MR participates in tolerance and coping strategies with aversive stimuli as well as arousal, wakefulness and long-term memory.

In humans, the MRF is activated during the transition from a relaxed awake state to an attention-demanding state during reaction-time tasks [46] and during the focused investigation of threats while interpreting the proximity of danger (e.g., freezing in place) [55]. The caudal portion of the MRF extends into the cuneiform region, which is correlated with cardiovascular dysfunction in ME/CFS [56,57] and pontis oralis (PO).

The right locus coeruleus (R_LC) had significantly lower activation in GWI than ME/CFS subjects after exercise [58,59]. This is the predominant source of noradrenergic innervation in the brain [58,59]. Integrated PAG, amygdala and sensory information activate the LC to generate diffuse efferent outputs to the cerebrum and brainstem [60]. They act in an instantaneous fashion like a tripwire for immediate instinctual responses such as freeze–fight–flight, focused cerebral attention and sympathetic activation for immediate action. Inappropriate or dysfunctional activation contributes to anxiety and PTSD [61]. Atrophy of the right locus coeruleus was found at autopsy in veterans with PTSD [62].

Exercise caused no incremental changes in controls (∆BOLD). However, exertion led to significant positive increments in ME/CFS and, by contrast, decreases in GWI. The dynamic changes were significantly different between ME/CFS and GWI for midline PAG, DR, MR and VTA, right LC, and bilateral PTN, PBC, and PO (Figure 4, Table 4). The opposite directions of change indicated that distinctly different pathological mechanisms that regulate midbrain blood flow and neurovascular coupling were modulated by exercise in the two diseases, indicating that ME/CFS and GWI were excellent “illness controls” for each other.

Both ME/CFS and GWI had significant light and sound sensitivity (Figure 4). These sensations are monitored in the superior and inferior colliculus, respectively. The inferior colliculi are innervated by the ascending hindbrain auditory pathway (the lateral lemniscus), somatosensory pathways from the medulla, pons, and arousal nuclei [63]. The inferior colliculus participates in multimodal sensory perceptions, vestibulo–ocular reflex, predator aversion and escape, prey localization, social communication, analgesia and fear-related behaviors. Sharp acoustic stimulation initiates the startle response with acutely accentuated attention and surveillance, leading to visual and truncal orientation towards the sound, generalized hyperarousal and aversive behaviors [64,65,66,67]. Inferior colliculus is relevant to the light and sound sensitivity and heightened startle response in ME/CFS, GWI and veterans with PTSD [68].

The inferior colliculus is highly metabolically active and vulnerable to toxic injury [63]. ME/CFS have reduced cerebral blood flow during heads up tilt [69] and exercise [70], which may reduce the oxygen supply to the susceptible inferior colliculus and lead to dorsal midbrain dysfunction. The exercise-induced lability of cerebral blood flow and neurovascular coupling may contribute to the dysfunctional BOLD patterns in the midbrain, insula and cerebellum vermis in the GWI and ME/CFS groups [14] but with different mechanisms and statistically reciprocal outcomes in the two diseases.

The general linear models assessed the relative influences and interactions between disease, orthostatic and gender status on each day and adjusted for age, PTSD, BMI and dolorimetry thresholds. Outcomes for disease status were generally comparable to the ANOVA results.

Orthostatic status was significant before exercise as START and POTS had higher BOLD than STOPP in the left PBC (Appendix A). POTS had postural tachycardia before and after exercise, while START were defined by exercise-induced postural tachycardia [26,27,31]. After exercise, START underwent a dynamic incremental depression of BOLD activation in the R_LC and bilateral PBC (Table 4) compared to the STOPP subjects who had no changes in BOLD or postural heart rate. The involvement of the locus coeruleus in START implies exercise-induced autonomic dysfunction. The PBC interrogates pain and interoceptive visceral sensations then forwards the information to the PAG, thalamus, hypothalamus, and amygdala for further processing [71]. PBC is recruited in states of malaise as an adaptive component of the sickness response [72] and so may participate in the experience of postexertional malaise.

Hedges’ g ranged from 0.50 to 0.81 for differences between groups by ANOVA (Table 3, Table 4 and Table 5), which suggests moderate-to-high effect sizes for replication of the significant results using the same protocols.

It is important to appreciate the limitations of these disease and exercise effects. The findings were inferred based on the seed regions extracted from the ascending arousal network. Coordinates of the seed regions may improve as newer standards are created [40]. The actual metric being compared is the 2-back>0-back differential activity during the difficult cognitive working memory task. The results may not be applicable to the resting state or other cognitive tasks. The neural and vascular responses combined to generate these data without providing insights into functional connectivity or molecular mechanisms. Brainstem motion may blur the borders of nuclei in this seed region approach. Therefore, we consider the results to be a general predictor of changes in BOLD for nuclei in the ascending arousal network that are congruent with the dorsal midbrain region of interest found in our previous study [14]. The results do suggest that significant differences will be found in future studies that specifically target these nuclei in ME/CFS and GWI when suitable sample sizes are compared and advanced motion correction algorithms are applied [73,74,75,76]. The most significant differences were induced by exercise with elevated BOLD in ME/CFS subjects but reductions in the GWI group, and were most clearly exposed by a comparison of ME/CFS vs. GWI groups rather than differences from control subjects (Table 5 and Appendix A).

## 5. Conclusions

The seed region approach based on the ascending arousal network extended our previous finding of exercise-induced changes in BOLD during a high-cognitive-load 2-back working memory task. The salient findings were significantly lower BOLD in the midbrain at baseline in the ME/CFS group compared to the GWI and control groups, and significant dynamic changes after exercise with elevation of BOLD in ME/CFS subjects but a reduction in the GWI group. A review of the functions of midbrain nuclei provides a fresh perspective on potential neural pathologies affecting inferior colliculus (startle), oculomotor and visual systems, PAG, MRF and other nuclei for threat assessment, anxiety, negative emotion, pain and tenderness and other aspects of the ME/CFS and GWI clinical experiences. The data provide an initial framework to power future studies of postexertional malaise and midbrain dysfunction.

## Figures and Tables

**Figure 1 brainsci-12-00078-f001:**
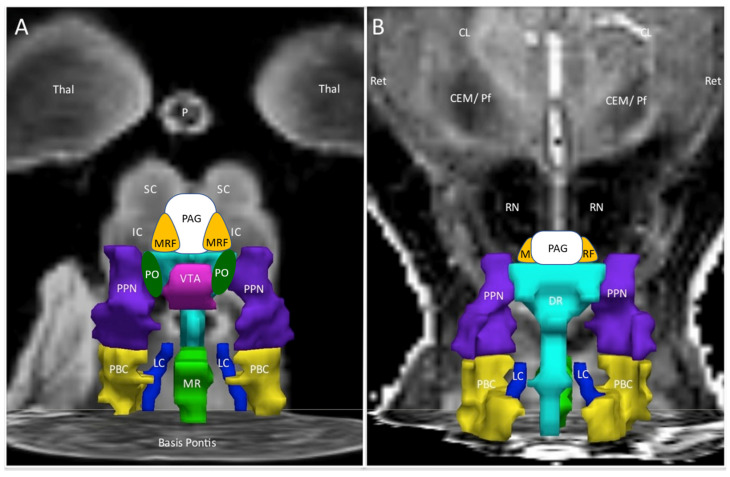
Ascending arousal network. The original figure [37] in the coronal plane was annotated by adding periaqueductal grey (PAG, white), midbrain reticular formation (MRF, orange) and pontis oralis (PO, dark green). (**A**) The anterior view showed the ventral midbrain with posterior thalamus (Thal) as a “ceiling”, the superior (SC) and inferior colliculi (IC) as the backdrop, and Basis Pontis as the “floor”. (**B**) The posterior view of the dorsal midbrain was oriented to show the centromedian/parafascicular nucleus (CEM/Pf), reticular nucleus (Ret), and central lateral nucleus (CL) of the thalamus, pineal (P) and midbrain red nuclei (RN). This depiction suggested three layers with the MRF and PAG being most rostral. The middle layer contained the ventral tegmental area (VTA, violet), bilateral pontis oralis (PO, dark green) and pedunculotegmental nuclei (L and R PTN, formerly pedunculopontine nuclei and labeled PPN in the original image, navy blue) and dorsal raphe (cyan) in the posterior midline. The caudal layer had median raphe (MR, green) and DR flanked by bilateral locus coeruleus (LC, navy blue) and parabrachial complex (PBC, yellow).

**Figure 2 brainsci-12-00078-f002:**
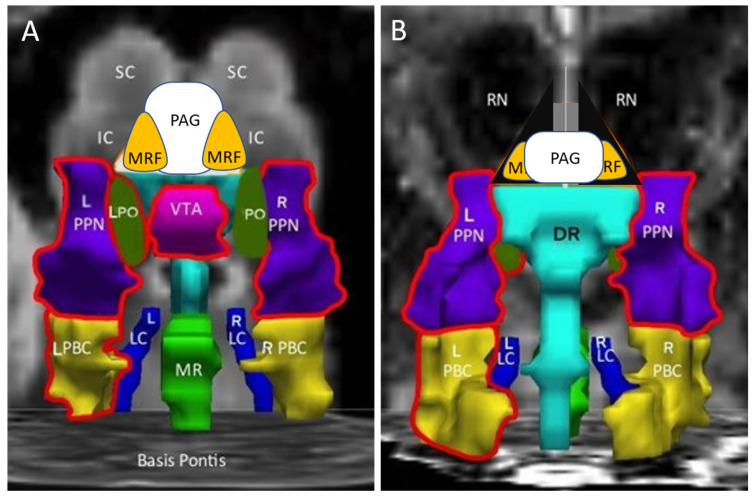
Ascending arousal network before exercise. Nuclei with significantly different levels of activation are highlighted with red in the (**A**) anterior/ventral and (**B**) posterior/dorsal views. Controls had greater activation than ME/CFS subjects with regard to ventral tegmental area (VTA), bilateral pedunculotegmental nuclei (L and R PTN, formerly pedunculopontine nuclei) and left parabrachial complex (L_PBC) by ANOVA with Tukey Honest Significant Difference *p* < 0.05 (Table 3).

**Figure 3 brainsci-12-00078-f003:**
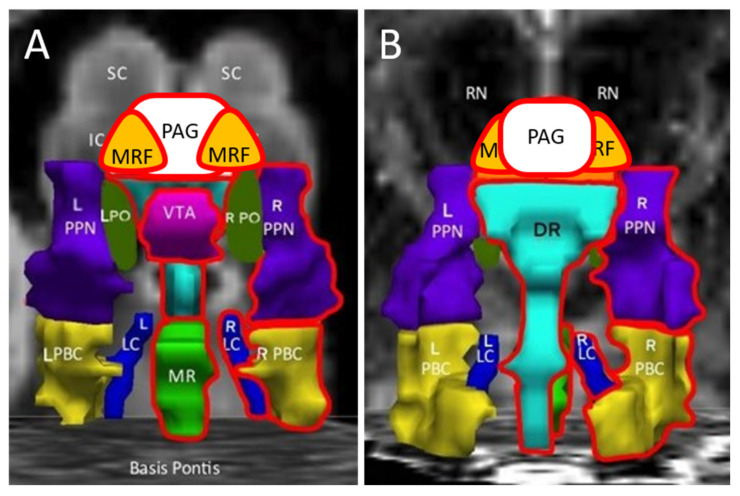
Post exercise. Anterior (**A**) and posterior (**B**) views indicate that GWI had significantly lower BOLD signals than control in the L_MRF, R_MRF, VTA and R_PTN (outlined in red, Tukey Honest Significant Difference, *p* < 0.05) (Table 4). GWI was significantly lower in the R_MRF, PAG, DR, MR, R_PBC and R_ LC than ME/CFS. There were no significant differences between the control and ME/CFS groups following exercise. Data are annotated as in Figure 1.

**Figure 4 brainsci-12-00078-f004:**
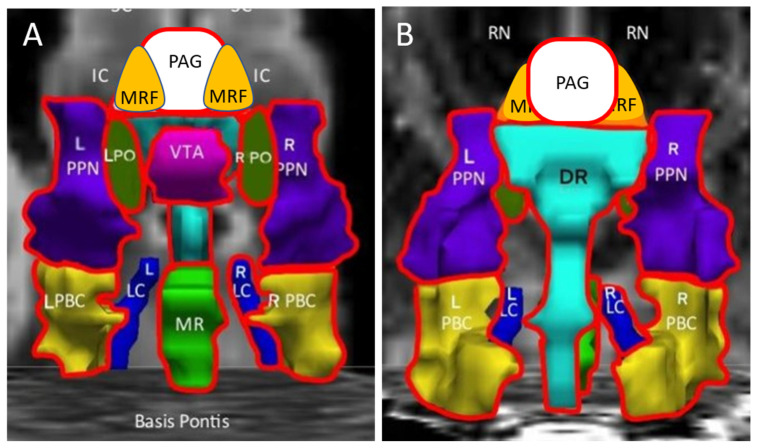
Exercise effects. Incremental changes (ΔBOLD) were significantly larger (more positive) for ME/CFS than GWI (negative, diminished BOLD) in the midline PAG, VTA, DR and MR, bilateral PBC, PO and PTN, and R_LC (outlined in red) in the anterior (**A**) and posterior (**B**) views. Controls had no net changes. Data are from ANOVA analysis (Table 5).

**Figure 5 brainsci-12-00078-f005:**
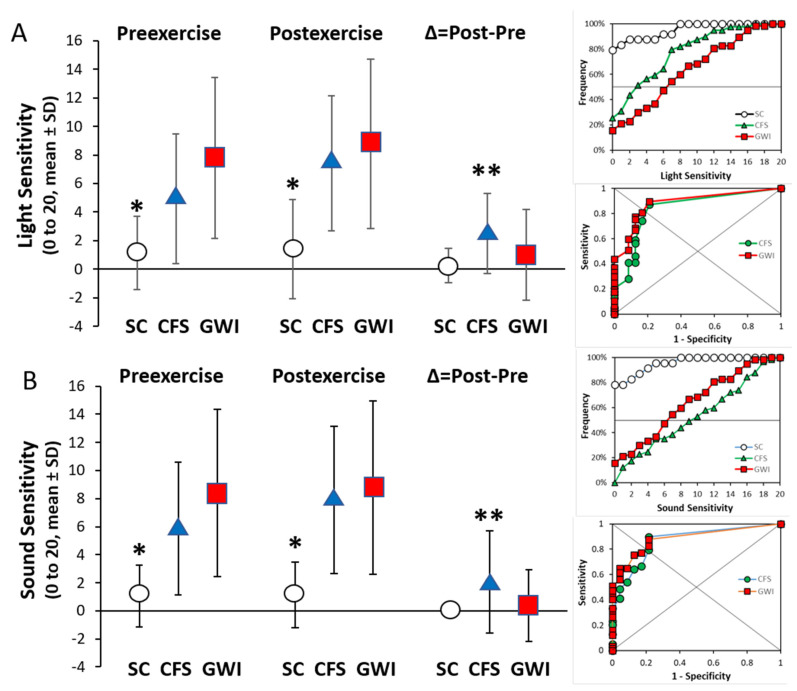
Light and sound sensitivity. Sensitivity was scored using modified Gracely box scores (range 0 to 20 points, mean ± SD). Light (**A**) and sound (**B**) were significantly worse for ME/CFS (blue triangles) and GWI (red squares) than SC (white circles) before and after exercise (* *p* < 0.0001 by 2-tailed unpaired *t*-test after Bonferroni correction). The incremental changes (∆ = Post minus Pre) were larger in ME/CFS than SC and GWI (** *p* < 0.044 by 2-tailed unpaired *t*-tests after Bonferroni correction). Paired changes were significant for light and sound in the ME/CFS group, and light in GWI. Receiver operating characteristics (depicted on the right) defined light sensitivity as ≥2 with 83.3% specificity and sensitivities of 74.4% for CFS and 80.7% for GWI (AUC = 0.884). Sound sensitivity of ≥2 had 78.3% specificity and sensitivities of 79.5% for ME/CFS and 82.5% for GWI (AUC = 0.884).

**Table 1 brainsci-12-00078-t001:** Abbreviations of nuclei in figures.

Abbreviation	Nuclei	Color
Thal CEM/Pf Ret CL P RN	Background nuclei for orientation thalamus centromedian/parafascicular nucleus reticular nucleus central lateral nucleus of the thalamus pineal midbrain red nuclei Basis Pontis—arbitrary section through the rostral pons to form the “floor” of the figure	grey scale
	Midbrain and Isthmus (superior to inferior)	
SC	superior colliculus	grey
IC	inferior colliculus	grey
PAG	periaquequctal grey	white
MRF	midbrain reticular formation	orange
VTA	ventral tegmental area	violet
DR	dorsal raphe	cyan
MR	median raphe	lime green
PTN	pedunculotegmental nuclei (PTN, formerly PPN pedunculopontine nuclei)	navy blue
PO	pontis oralis	dark green
PBC	parabrachial complex	yellow
LC	locus coeruleus	royal blue

**Table 2 brainsci-12-00078-t002:** Demographics. Questionnaire scores were significantly different between control, ME/CFS and GWI by ANOVA followed by Tukey Honest Significant Difference or Kruskal–Wallis and Mann–Whitney tests (*) corrected for multiple comparisons. Mean ± SD. (Score ranges).

	Control	ME/CFS	GWI	Control vs. ME/CFS	Control vs. GWI	ME/CFS vs. GWI
N	31	36	78			
Age	43.2 ± 16.5	47.3 ± 13.1	47.1 ± 7.4			
Female *	38.7%	69.4%	21.8%	0.029		0.0001
BMI	28.3 ± 4.5	26.2 ± 5.6	29.4 ± 5.3			0.008
Dolorimetry (kg)	4.6 ± 2.4	3.9 ± 1.9	4.2 ± 2.1			
PTSD *	9.7%	13.8%	43.6%		0.001	0.004
Migraine *	13.3%	41.7%	63.5%		<0.00001	
Chalder Fatigue (0 to 33)	12.3 ± 5.4	23.3 ± 6.2	25.3 ± 4.7	<0.00001	<0.00001	
McGill pain (0 to 45)	3.6 ± 6.4	13.4 ± 11.0	23.8 ± 9.0	0.00012	<0.00001	<0.00001
CFS Severity Questionnaire (0 no symptom to 4 severe)						
Fatigue	1.3 ± 1.2	3.4 ± 0.8	3.5 ± 0.7	<0.00001	<0.00001	
Postexertional Malaise	0.6 ± 1.1	3.5 ± 0.8	3.4 ± 1.0	<0.00001	<0.00001	
Sleep	1.7 ± 1.4	3.2 ± 0.9	3.5 ± 0.8	<0.00001	<0.00001	
Memory, Concentration	1.2 ± 1.2	2.9 ± 0.9	3.1 ± 0.8	<0.00001	<0.00001	
Muscle Pain	0.6 ± 0.9	2.5 ± 1.3	3.1 ± 1.0	<0.00001	<0.00001	0.011
Joint Pain	0.8 ± 1.0	1.9 ± 1.4	3.2 ± 1.0	0.00026	<0.00001	<0.00001
Headache	1.0 ± 1.2	2.0 ± 1.3	2.7 ± 1.2	0.0022	<0.00001	0.025
Sore Throat	0.3 ± 0.7	1.0 ± 1.0	1.4 ± 1.2	0.031	0.000013	
Lymph Nodes	0.1 ± 0.4	1.0 ± 1.1	1.5 ± 1.3	0.004	<0.00001	
SF-36 (100 best to 0 worst)						
Physical Function	85.2 ± 24.2	44.4 ± 26.5	46.9 ± 24.6	<0.00001	<0.00001	
Role Physical	80.0 ± 36.8	10.0 ± 25.9	9.5 ± 24.8	<0.00001	<0.00001	
Bodily Pain	82.9 ± 20.1	47.2 ± 27.7	29.5 ± 18.3	<0.00001	<0.00001	0.00024
General Health	69.8 ± 22.8	33.2 ± 22.9	26.4 ± 19.2	<0.00001	<0.00001	
Vitality	60.2 ± 20.7	18.0 ± 15.9	16.6 ± 15.3	<0.00001	<0.00001	
Social Function	80.0 ± 25.1	30.7 ± 27.0	30.8 ± 24.5	<0.00001	<0.00001	
Role Emotional	86.7 ± 31.1	70.5 ± 44.1	30.7 ± 38.4		<0.00001	<0.00001
Mental Health	73.6 ± 16.8	67.8 ± 17.4	54.8 ± 22.3		0.000084	0.0056

**Table 3 brainsci-12-00078-t003:** ANOVA for pre-exercise differences based on disease status. BOLD signals for SC and GWI were numerically and statistically higher than ME/CFS (mean ± 95%CI, Tukey Honest Significant Difference, HSD). L and R indicate left and right, respectively. Hedges’ g.

	Control	ME/CFS	GWI	Control > ME/CFS	GWI > ME/CFS
L_MRF	0.266 ± 0.164	0.123 ± 0.124	0.197 ± 0.103		
R_MRF	0.331 ± 0.181	0.216 ± 0.119	0.172 ± 0.103		
PAG	0.234 ± 0.145	0.061 ± 0.101	0.222 ± 0.089		
VTA	0.279 ± 0.127	0.103 ± 0.089	0.193 ± 0.061	*p* = 0.034, g = 0.58	
DR	0.275 ± 0.118	0.086 ± 0.103	0.233 ± 0.081		
MR	0.295 ± 0.124	0.131 ± 0.123	0.249 ± 0.087		
L_PO	0.263 ± 0.134	0.061 ± 0.126	0.257 ± 0.078		*p* = 0.019, g = 0.56
R_PO	0.290 ± 0.134	0.103 ± 0.136	0.252 ± 0.087		
L_PTN	0.316 ± 0.133	0.049 ± 0.101	0.264 ± 0.086	*p* = 0.008, g = 0.81	*p* = 0.009, g = 0.60
R_PTN	0.356 ± 0.126	0.125 ± 0.086	0.259 ± 0.080	*p* = 0.014, g = 0.78	
L_LC	0.309 ± 0.147	0.142 ± 0.158	0.226 ± 0.103		
R_LC	0.316 ± 0.181	0.143 ± 0.175	0.274 ± 0.126		
L_PBC	0.331 ± 0.139	0.078 ± 0.135	0.231 ± 0.092	*p* = 0.029, g = 0.65	
R_PBC	0.299 ± 0.148	0.088 ± 0.136	0.263 ± 0.101		

**Table 4 brainsci-12-00078-t004:** ANOVA for postexercise by disease status. Control and ME/CFS had significantly higher BOLD than GWI for several nodes. There was a general trend for GWI to have the lowest values in all regions. Mean ± 95%CI. Tukey Honest Significant Difference. Hedges’ g.

	Control	ME/CFS	GWI	Control > GWI	ME/CFS > GWI
N	31	36	78		
L_MRF	0.341 ± 0.146	0.206 ± 0.147	0.073 ± 0.094	*p* = 0.008, g = 0.65	
R_MRF	0.364 ± 0.149	0.301 ± 0.138	0.077 ± 0.084	*p* = 0.002, g = 0.75	*p* = 0.014, g = 0.58
PAG	0.182 ± 0.141	0.271 ± 0.142	0.060 ± 0.078		*p* = 0.016, g = 0.57
VTA	0.252 ± 0.104	0.195 ± 0.099	0.078 ± 0.068	*p* = 0.018, g = 0.58	
DR	0.229 ± 0.113	0.265 ± 0.126	0.068 ± 0.070		*p* = 0.009, g = 0.60
MR	0.231 ± 0.115	0.285 ± 0.139	0.092 ± 0.075		*p* = 0.018, g = 0.54
L_PO	0.178 ± 0.125	0.241 ± 0.141	0.129 ± 0.082		
R_PO	0.248 ± 0.131	0.255 ± 0.135	0.101 ± 0.077		
L_PTN	0.233 ± 0.124	0.208 ± 0.127	0.093 ± 0.084		
R_PTN	0.308 ± 0.158	0.289 ± 0.104	0.126 ± 0.074	*p* = 0.04, g = 0.50	
L_LC	0.223 ± 0.144	0.274 ± 0.146	0.097 ± 0.089		
R_LC	0.276 ± 0.167	0.313 ± 0.150	0.077 ± 0.095		*p* = 0.022, g = 0.55
L_PBC	0.250 ± 0.133	0.250 ± 0.135	0.106 ± 0.091		
R_PBC	0.240 ± 0.135	0.289 ± 0.139	0.110 ± 0.076		*p* = 0.041, g = 0.50

**Table 5 brainsci-12-00078-t005:** ΔBOLD. Exercise-induced changes in the 2-back>0-back condition were analyzed. Incremental changes (ΔBOLD = postexercise minus pre-exercise) were assessed by ANOVA then corrected for multiple comparisons with Tukey Honest Significant Difference. ME/CFS subjects had increased BOLD, while the GWI group had diminished BOLD after exercise. Control had no net change and was not statistically different from either of the other groups. Mean ± 95%CI. Hedges’ g.

	SC	ME/CFS	GWI	ME/CFS > GWI
N	31	36	78	
L_MRFΔ	0.075 ± 0.201	0.083 ± 0.163	−0.124 ± 0.129	
R_MRFΔ	0.033 ± 0.216	0.085 ± 0.171	−0.095 ± 0.122	
PAGΔ	−0.052 ± 0.167	0.210 ± 0.175	−0.162 ± 0.110	*p* = 0.001, g = 0.75
VTAΔ	−0.027 ± 0.138	0.092 ± 0.130	−0.114 ± 0.080	*p* = 0.016, g = 0.57
DRΔ	−0.046 ± 0.139	0.179 ± 0.159	−0.165 ± 0.103	*p* = 0.001, g = 0.75
MRΔ	−0.065 ± 0.153	0.155 ± 0.173	−0.157 ± 0.113	*p* = 0.005, g = 0.62
L_POΔ	−0.085 ± 0.167	0.180 ± 0.181	−0.128 ± 0.108	*p* = 0.006, g = 0.62
R_POΔ	−0.041 ± 0.178	0.152 ± 0.175	−0.151 ± 0.123	*p* = 0.014, g = 0.56
L_PTNΔ	−0.083 ± 0.154	0.158 ± 0.166	−0.171 ± 0.107	*p* = 0.002, g = 0.69
R_PTNΔ	−0.048 ± 0.168	0.164 ± 0.155	−0.132 ± 0.100	*p* = 0.004, g = 0.66
L_LCΔ	−0.086 ± 0.172	0.133 ± 0.201	−0.129 ± 0.130	
R_LCΔ	−0.039 ± 0.218	0.169 ± 0.220	−0.197 ± 0.165	*p* = 0.024, g = 0.52
L_PBCΔ	−0.081 ± 0.181	0.171 ± 0.198	−0.125 ± 0.128	*p* = 0.025, g = 0.52
R_PBCΔ	−0.059 ± 0.167	0.202 ± 0.195	−0.153 ± 0.129	*p* = 0.005, g = 0.62

**Table 6 brainsci-12-00078-t006:** Multivariate general linear model mGLM for ΔBOLD and orthostatic status. Estimated marginal means for incremental changes (ΔBOLD) were evaluated with disease and orthostatic status, PTSD and gender as fixed factors and age, BMI and dolorimetry as independent variables. The model was significant for disease status (Appendix A) and orthostatic status (Wilks’ Lambda = 0.651, *p* = 0.019, Partial Eta Squared = 0.193) but not age, gender, PTSD, BMI or dolorimetry. Additionally, 95% confidence intervals for POTS and STOPP bracketed zero, indicating no net changes with exercise. The START phenotype was defined by exercise-induced postural tachycardia and had reduced BOLD activation in the R_LC and bilateral PBC following exercise. The STOPP phenotype was the normal condition with no change in postural tachycardia. Mean ± 95%CI. Univariate significance. Hedges’ g.

	POTS	START	STOPP	STOPP > START
L_MRFΔ	−0.009 ± 0.315	−0.093 ± 0.235	−0.063 ± 0.174	
R_MRFΔ	0.003 ± 0.319	−0.096 ± 0.239	−0.051 ± 0.177	
PAGΔ	0.004 ± 0.271	−0.140 ± 0.202	−0.026 ± 0.150	
VTAΔ	−0.062 ± 0.211	−0.146 ± 0.157	0.034 ± 0.116	
DRΔ	−0.090 ± 0.251	−0.153 ± 0.188	0.028 ± 0.139	
MRΔ	−0.164 ± 0.271	−0.166 ± 0.203	0.050 ± 0.150	
L_POΔ	−0.085 ± 0.274	−0.170 ± 0.204	0.061 ± 0.151	
R_POΔ	−0.048 ± 0.298	−0.259 ± 0.222	0.072 ± 0.164	
L_PTNΔ	−0.247 ± 0.256	−0.098 ± 0.191	−0.028 ± 0.142	
R_PTNΔ	−0.053 ± 0.260	−0.098 ± 0.194	−0.008 ± 0.144	
L_LCΔ	−0.219 ± 0.314	−0.232 ± 0.235	0.102 ± 0.174	
R_LCΔ	−0.070 ± 0.390	−0.329 ± 0.291	0.156 ± 0.216	*p* = 0.035, g = 0.42
L_PBCΔ	−0.304 ± 0.302	−0.268 ± 0.226	0.117 ± 0.167	*p* = 0.029, g = 0.29
R_PBCΔ	0.020 ± 0.316	−0.298 ± 0.236	0.110 ± 0.174	*p* = 0.026, g = 0.39

## Data Availability

Primary data are attached in Appendix A.

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
