# Peer review of "Differential Effects of Exercise on fMRI of the Midbrain Ascending Arousal Network Nuclei in Myalgic Encephalomyelitis/Chronic Fatigue Syndrome (ME/CFS) and Gulf War Illness (GWI) in a Model of Postexertional Malaise (PEM)"

_brainsci, 2022, doi:10.3390/brainsci12010078_

Round 1

Reviewer 1 Report

Visual Abstract: Green overlap of PAG (white) confusing. Should it be 'DR dorsal raphe (light green)' ?

line 34: 'unrefreshing sleep and total body pain' sufficient?

line 38: after 'criteria' add '(known as Fukuda criteria)'

Should 'SOM Figure S1' be omitted in manuscript?

line 170 refers to (Fig. 1) which does not exist? A timing of events figure would be helpful.

line 180: '... a pooled populations grand mean ...' to be clear.

line 199: 'penduclo...' omit 'n'.

Please add table listing midbrain nuclei, their abbreviations and volumes (voxels). Repeatedly slogging through Figure 1 caption to find abbreviations is hard work! Can then simplify Figure 1 caption. Lead captions with "Red bordered regions had significantly lower BOLD ....".  'Basis Pontis' uninformative.  'level of pontomedullary sulcus (z = -36)' or similar would be better?

Improve quality of Fig 1 to that of Figs 2 and 3.

line 286 :.. views indicate with red borders where GWI had ...'

line 289 'Structures are annotated as in table 1'. - (referring to new table)

line 201: '... but embryologically extends ...'

line 372: '... is that it can reveal differences in relative levels ...'

Because studies of brainstem ascending reticular activation system nuclei is so novel and the ARAS is so unfamiliar, it is important to clearly introduce the nuclei you study. To this end a figure like S1 without the red result areas and with larger (one-sided) sagittal images would be desirable before showing in result figures the red voxels of difference and the red-bordered regions of difference.

Reviewer 2 Report

This manuscript describes a re-analysis of data from an earlier study with 3 comparison groups: individuals with Myalgic Enchephalomyilitis/Chronic Fatigue Syndrome (ME/CFS) or with Gulf War Illness (GWI), and a non-diseased control group. The original study consisted of cognitive tests being performed by study participants from all 3 groups during fMRIs, before and after submaximal exercise. The re-analysis reported in the paper is focused on a seed region approach and the ascending arousal network (Harvard Ascending Arousal Network).

The manuscript contains relevant information related to the original study and the justification for the re-analysis approach.  The methods and results sections are well described, and my only suggestion is to replace the titles of the tables and figures with a shorter descriptive title, and include the explanations, which are currently in the titles, in the main text. Additionally, all acronyms presented in each figure and table, should be spelled out in spelled out in endnotes, unless the authors have reached the word count requested by the journal.

This is a well thought and analysed study, quite with intriguing results, particularly those related to the opposite effects to exercise found between the GWI and ME/CFS groups. The authors rightly acknowledged the limitations of their findings, but point to the potential use of their data as a framework for future studies on post-exertion malaise, focusing on the midbrain region.
